# A New Paradigm in the Relationship between Melatonin and Breast Cancer: Gut Microbiota Identified as a Potential Regulatory Agent

**DOI:** 10.3390/cancers13133141

**Published:** 2021-06-23

**Authors:** Aurora Laborda-Illanes, Lidia Sánchez-Alcoholado, Soukaina Boutriq, Isaac Plaza-Andrades, Jesús Peralta-Linero, Emilio Alba, Alicia González-González, María Isabel Queipo-Ortuño

**Affiliations:** 1Unidad de Gestión Clínica Intercentros de Oncología Médica, Hospitales Universitarios Regional y Virgen de la Victoria, Instituto de Investigación Biomédica de Málaga (IBIMA)-CIMES-UMA, 29010 Málaga, Spain; aurora.laborda@ibima.eu (A.L.-I.); l.sanchez.alcoholado@ibima.eu (L.S.-A.); soukaina@ibima.eu (S.B.); isaac.plaza.andrades@ibima.eu (I.P.-A.); jesus.peralta@ibima.eu (J.P.-L.); maribel.queipo@ibima.eu (M.I.Q.-O.); 2Facultad de Medicina, Universidad de Málaga, 29071 Málaga, Spain; 3Centro de Investigación Biomédica en Red de Cáncer (Ciberonc CB16/12/00481), 28029 Madrid, Spain

**Keywords:** melatonin, breast cancer, gut microbiota, circadian disruption, dysbiosis, short-chain fatty acids, estrogens, estrobolome, anticancer therapies, tryptophan metabolism

## Abstract

**Simple Summary:**

The relationship between melatonin and breast cancer has been widely described. On the other hand, in recent years, an imbalance in the composition of the intestinal bacterial population has been linked as another possible trigger for this disease. Given that changes in the gut microbiota have been observed to stimulate the kinurenine pathway, reducing circulating melatonin levels, in this review, we summarize the relationship between circadian disruption and breast cancer, as well as the connection with dysbiosis as possible causing this pathology due to a series of changes that lead to an increase in circulating estrogen levels.

**Abstract:**

In this review we summarize a possible connection between gut microbiota, melatonin production, and breast cancer. An imbalance in gut bacterial population composition (dysbiosis), or changes in the production of melatonin (circadian disruption) alters estrogen levels. On the one hand, this may be due to the bacterial composition of estrobolome, since bacteria with β-glucuronidase activity favour estrogens in a deconjugated state, which may ultimately lead to pathologies, including breast cancer. On the other hand, it has been shown that these changes in intestinal microbiota stimulate the kynurenine pathway, moving tryptophan away from the melatonergic pathway, thereby reducing circulating melatonin levels. Due to the fact that melatonin has antiestrogenic properties, it affects active and inactive estrogen levels. These changes increase the risk of developing breast cancer. Additionally, melatonin stimulates the differentiation of preadipocytes into adipocytes, which have low estrogen levels due to the fact that adipocytes do not express aromatase. Consequently, melatonin also reduces the risk of breast cancer. However, more studies are needed to determine the relationship between microbiota, melatonin, and breast cancer, in addition to clinical trials to confirm the sensitizing effects of melatonin to chemotherapy and radiotherapy, and its ability to ameliorate or prevent the side effects of these therapies.

## 1. Introduction

Melatonin, or *N*-acetyl-5-methoxytryptamine, is the main hormone secreted by the pineal gland (a photoneuroendocrine transducer that converts light information into humoral signals) [1,2], and its synthesis is controlled by suprachiasmatic nuclei in the hypothalamus. This hormone presents a rhythmic production profile which is proportional to the nocturnal noradrenergic stimulus, with minimum diurnal values and maximum nocturnal values.

Melatonin, apart from being synthesized in the pineal gland at night, is also produced in other extrapineal sites such as the retina, the gastrointestinal tract, skin, bone marrow, and lymphocytes, where it can act as an intracellular mediator or paracrine signal, in addition to having endocrine effects [3]. In particular, it is found in vast amounts in intestinal cells. Moreover, it should be noted that intestinal microbiota directly or indirectly participate in the production of this hormone. On the one hand, microbial metabolism produces melatonin directly [4], and, on the other hand, gut bacteria indirectly produce short-chain fatty acids (SCFAs) that stimulate the production of serotonin, which is then converted into melatonin through the actions of arylalkylamine-*N*-acetyltransferase (AANAT) and acetylserotonin O-methyltransferase (ASMT), through the melatonergic pathway (Figure 1).

Melatonin formation begins with the uptake of tryptophan (Trp), coming from the bloodstream through its melatonergic pathway [2]. However, in addition to the Trp melatonergic pathway, there is another alternative pathway, the kynurenine pathway, which is implicated in the development of breast cancer. Breast cancer patients present an increase in pro-inflammatory cytokines (IL-1β, IL-6, IL-18, TNFα and IFNγ) [5]. These pro-inflammatory cytokines induce the synthesis of the extrahepatic enzyme IDO, which removes Trp from serotonin synthesis, favouring the activation of the kynurenine pathway, and therefore increases the production of the aryl hydrocarbon receptor (AhR) ligands kynurenine and kynurenic acid [6]. The activation of this receptor is particularly relevant in breast cancer [7], and high levels of its ligands are associated with breast cancer, especially triple-negative breast cancer [8]. High concentrations of indoleamine 2,3-dioxygenase (IDO) are associated with poorer survival [8] and increased angiogenesis and metastasis in breast cancer [9] (see Figure 1). Furthermore, an increase in melatonin metabolism is observed in these patients, leading to a reduction in its circulating levels [8].

In contrast, oxidative stress induces high levels of the liver enzyme tryptophan 2,3-dioxygenase (TDO), which also leads tryptophan to the kynurenine pathway and therefore to the activation of the AhR receptor [7]. Once this receptor is activated, AhR/CYP1B1 signalling begins, which has implications for cancer cell survival and proliferation, disease progression, and chemoresistance. The activation of this pathway implies an increase of *N*-acetylserotonin (NAS) in the NAS/melatonin ratio (see Figure 1). The induction of NAS mediated by CYP1B1 in these cells activates tyrosine kinase B receptors (TrkB) that contributes to the survival and migration of breast cancer cells and prevents the effects of melatonin in the mitochondria [6]. In parallel, a reduction of the pineal hormone in mammary tumour cells results in the AhR effects [7].

In turn, these alterations in the melatonergic pathways which occur in breast cancer cells generate changes in microbiome and intestinal permeability. These variations, particularly the reduction in butyrate levels and the increase in bacterial lipopolysaccharides (LPS), lead to an increase in NOS and oxidative stress, a strong autoimmune pro-inflammatory response, and the production of Trp catabolites, thus affecting tryptophan metabolism and melatonin production [6]. Furthermore, Kassayova et al. corraborated this connection between melatonin and gut microbiota, describing how *Lactobacillus* and inulin exert anti-inflammatory, antiproliferative, immunomodulatory, and prodifferentiating activities, which are enhanced when administered in combination with melatonin, thus indicating the possible relationship that melatonin and the intestinal microbiota exert on breast cancer risk [10].

Therefore, a bidirectional relationship is seen, in which changes in intestinal microbiota influence the regulation of the melatonergic pathway, and alterations in melatonin synthesis generate changes in microbiota, with both factors being able to contribute to the pathoetiology of breast cancer and its treatment.

This review aims to describe the bases on which the use of melatonin as an adjunctive therapy in breast cancer is supported, principally relating to its antiestrogenic properties. On the other hand, the possible relationship between circadian disruption, due to alterations in melatonin levels, and the intestinal dysbiosis, caused by an imbalance in the bacterial composition of estrobolome is also highlighted, as this leads to an increase in estrogen levels which promotes the appearance of breast cancer. Finally, a section on clinical trials investigating melatonin use in breast cancer is presented.

## 2. Search Methods

In order to collate publications analyzing the relationship between melatonin, gut microbiota and breast cancer, a literature review was done using PubMed, Scopus, EMBASE, and the Web of Science databases. To perform the search we conducted a free-text search using these keywords: melatonin, breast cancer, microbiota, microbiome, dysbiosis, gut, intestinal homeostasis, and estrobolome, as well as thesaurus descriptors search using MeSH and Emtree (adapted for the selected databases). The inclusion criteria for eligible articles were “Publication in peer-reviewed journals up to May 2021”, and “The English language”. Articles were excluded by abstract or full text due to their irrelevance to the analyzed topic. Finally, the references of the selected articles were also reviewed to identify any other studies that met the inclusion criteria.

## 3. Melatonin Interaction Molecules

Melatonin’s actions depend on its binding to specific receptors in target tissues. Two basic types of melatonin receptors have been defined (Figure 2). The first type consists of cell membrane receptors (MT1, MT2, MT3) coupled to guanosine triphosphate (GTP)-binding proteins. MT1 and MT2 receptors suppress adenylyl cyclase activity via inhibitory G proteins sensitive to pertussis toxin, which leads to a reduction in intracellular levels of cAMP, resulting in a change in the phosphorylation state of target proteins [11]. MT1 is found in the Pars Tuberalis of the pituitary, and in the suprachiasmatic nuclei in the hypothalamus; it is encoded by the melatonin receptor 1A (*MTNR1A*) gene and is related to the circadian and reproductive functions of melatonin [12]. MT2, on the other hand, is found in the retina; is encoded by the melatonin receptor 1B (*MTNR1B*) gene, and is involved in the melatonin phase change response [12]. These two receptors are expressed in many tissues in the central nervous system (CNS), as well as in extraneural tissues, including mammary epithelial cells [12]. In particular, the possible role of MT1 in breast cancer has been investigated [13].

The second type of receptor belongs to the retinoid Z receptor/retinoid receptor-related orphan nuclear receptor alpha and beta (RZR/ROR-α and ROR-β) superfamily (Figure 2). Given the lipophilic nature of melatonin and its ability to easily cross the plasma membrane and reach the cell nucleus, the possibility that some of these receptors could be binding sites for this hormone has been considered, although this has not yet been verified [14].

Another mechanism of action of melatonin may not be mediated by receptors, since it may interact directly with a cytosolic protein called calmodulin (CaM) [11,15] (Figure 2). The binding of 3H-melatonin to calmodulin occurs due to its liposolubility, crossing the cell membrane and interacting directly with calmodulin [16]. This binding is specific, saturable, reversible, calcium dependent, ligand selective, and shows high affinity. This high binding affinity suggests that melatonin is capable of modulating many intracellular functions and that cellular activity may therefore depend on circulating melatonin levels [2,16]. Melatonin is an endogenous antagonist of calmodulin, inducing conformational changes in the ERα-CaM complex, thus preventing the binding of the E_2_-ERα-CaM complex to DNA and therefore preventing ERα transcription, in addition to decreasing the affinity of ER for estradiol [15].

Finally, several studies have described melatonin as a powerful free radical scavenger, due to its ability to transfer electrons to hydroxyl radicals, superoxide anions, hydrogen peroxide, hypochlorous acid, nitric oxide, and peroxynitric anions [17] (Figure 2). Furthermore, melatonin has the ability to stimulate the expression of antioxidant enzymes. All of this suggests that melatonin provides protection to cells from oxidative damage [2,11].

### 3.1. Melatonin Actions in Cancer

Melatonin is a hormone with different mechanisms of action which have been previously defined in various biological contexts. Primarily, melatonin, acting through the pineal gland, is the essential link for the synchronization of different circadian and circannual rhythms with ambient light. In humans, melatonin is able to synchronize the sleep-wake cycle in blind subjects, and to improve or alleviate the symptoms of the disorders resulting from transmeridian flights, commonly referred to as jet-lag. In addition, an increase in the incidence of breast cancer has been observed in women who work night shifts, as exposure to artificial light at night (ALAN) is related to lower melatonin production [18].

Secondly, a large body of work has described the antioxidant properties of melatonin through the neutralization of free radicals [12]. Considering that free radicals are involved in carcinogen-mediated DNA modifications, it has been suggested that melatonin could protect cells from the initiation of tumour processes. Moreover, it prevents nuclear DNA damage by counteracting reactive oxygen and nitrogen species [12]. Damaged DNA can undergo mutations and eventually cause malignant transformations. If this damage persists and is not repaired, it can continue to accumulate over an individual’s lifespan and is then likely to be one of the main causes of cancer in old age [12]. Melatonin has been reported to transfer electrons between antioxidant and pro-oxidant species [17], and has redox properties due to the presence of an electron-rich aromatic ring system, which allows this indoleamine to easily function as an electron donor [17]. Furthermore, due to its *O*-methyl and *N*-acetyl residues, melatonin is an amphiphilic compound.

Third, another action of melatonin is its immune-system modulation by exerting immunostimulant actions mediated by interleukins and other cytokines on monocytes and lymphocytes. This means that low levels of melatonin in serum could alter the immune system by reducing tumour surveillance and increasing tumour cell proliferation [2].

Finally, there are melatonin’s antitumour actions [1,12]. Among these, it is worth highlighting its actions on the hypothalamic-pituitary-gonad axis, underscoring its antiestrogenic nature. Furthermore, given this characteristic, melatonin can act as a selective estrogen receptor modulator (SERM), as well as a selective estrogen enzyme modulator (SEEM) [12].

Furthermore, melatonin has antiproliferative actions and induces apoptosis in tumour cells. These changes have been associated with the arrest of the cell cycle, by increasing the duration of the GAP1 (G_1_) cell growth phase, delaying entry into the DNA synthesis phase (S) and mitosis [19]. Melatonin stimulates apoptosis by increasing *p53*, which induces apoptosis by decreasing the expression of the B2 cell lymphoma gene (*Bcl2*), and increasing that of the Bcl2-associated X protein (Bax) as well as cyclin-dependent kinase inhibitor 1 (p21^WAF1^) [20]. Melatonin decreases apoptosis in immune-system cells, and in neurons in cases of immunodeficiency or neurodegeneration, while it increases apoptosis in cancer cells [11].

In fact, melatonin inhibits telomerase activity and reduces the growth of human mammary tumour cells [21]. This enzyme is essential for the synthesis of specialized ribonuclear proteins (telomeres) which extend the ends of eukaryotic linear chromosomes. Telomeres are vital to stabilizing chromosome structure, since with each cell division they become shorter, weakening the structure of the chromosome and leading to genetic instabilities and cell aging. Conversely, telomerase activity in cancer cells is overexpressed, maintaining the stability of DNA and thus contributing to its immortality, providing an unlimited capacity for the division of neoplastic cells [12]. Human telomerase reverse transcriptase (hTERT) is the subunit of telomerase that determines the main activity of this enzyme and therefore serves as an indicator of its activation. Melatonin inhibits estradiol- or cadmium-induced hTERT transcription in the MCF-7 breast cancer cell line, and reduces the trans-activation of hTERT initiated by ERα and mediated by estradiol or cadmium [21].

Apart from all these antitumoral actions, melatonin also inhibits invasion and migration, important mechanisms for metastasis [12,22]. Melatonin inhibits these processes by preventing tumour cells from entering the vascular system and stopping tumour angiogenesis from occurring by preventing the formation of secondary blood vessels at distant sites [23]. Furthermore, Borin et al. have described a mechanism by which increased expression of Rho-associated protein kinase (ROCK-1) is associated with tumour growth and metastasis in breast cancer, and this expression can be inhibited by melatonin [24].

Melatonin also acts on the metabolism of fats, limiting the adsorption of linoleic acid, a fatty acid that promotes tumour growth. This fatty acid, after its entry into the cell, initiates a series of events which culminate in cell proliferation. Some of these events involve epidermal growth factor (EGF), the phosphorylation and stimulation of signalling molecule cascades, and mitogen-activated kinases, MAPK kinase (MEK or MAPKK) and ERK1/2. Thus, melatonin modifies tumour growth by reducing the adsorption and metabolism of linoleic acid [12].

### 3.2. Melatonin as an Anti-Estrogen: SERM and SEEM Properties

Among the antitumor actions of melatonin, its ability to interact with the estrogen signalling pathway is crucial. The antiestrogenic effects of melatonin are explained on the one hand by its indirect actions on the neuroendocrine-reproductive axis, wherein melatonin, acting on the hypothalamus, pituitary, and the gonads, reduces the synthesis of ovarian estrogens and prolactin, which are hormones with important roles in both normal and tumour breast growth [2]. Melatonin decreases FSH and LH concentrations by acting on the hypothalamus, and inhibits prolactin synthesis, storage and secretion, which induces reduced gonadal steroids synthesis [25] (Figure 3).

On the other hand, melatonin also exerts direct antiestrogenic actions at the level of mammary tumour cells, interacting with estrogen receptors and counteracting the effects of estrogens, acting as a SERM [26]. Melatonin decreases the expression of ERα and inhibits the binding of the estradiol (E_2_)-ERα complex to the estrogen response element (ERE) in DNA, this effect being dependent on the prior binding of melatonin to its MT1 membrane receptor [15]. The binding of melatonin to its receptor causes a decrease in cAMP due to the inhibition of adenylate cyclase, which prevents the phosphorylation of ERα required for its binding to ERE and the initiation of transcription. However, other authors postulate that the binding of melatonin to calmodulin prevents the interaction of this protein with the E_2_-ERα complex, inhibiting binding to ERE, and therefore gene transcription [15]. It should be noted that melatonin, unlike other synthetic anti-estrogens such as tamoxifen, does not affect the binding of ERα coactivators [15]. Thus, it counteracts the effects of estradiol on its target tissues, acting as a natural anti-estrogen.

A third mechanism by which melatonin can reduce the development of estrogen-dependent tumours is based on its ability to modulate the activity of enzymes involved in the synthesis and transformation of estrogens in tumour tissue, with melatonin acting as a selective modulator for the enzymes involved in the local synthesis of estrogens (SEEM) [27]. Melatonin also inhibits the expression and activity of aromatase, which is involved in the conversion of androgens into estrogens [27]. This inhibition is accomplished through the inhibition of the expression of promoter regions of aromatase (promoters II, I.3 and I.4) [28]. The activation of these promoters is directly regulated by cAMP and by factors that intervene in the regulation of pathways that modify cAMP levels, such as prostaglandin E2. Thus, melatonin, through its inhibitory effect on the expression and activity of cyclooxygenases in breast cancer, decreases the production of prostaglandin E2, which reduces the levels of cAMP and indirectly decreases the activation of aromatase promoters II and I.3, decreasing aromatase expression and activity and therefore estrogen production [28]. Melatonin inhibits COX2, either by preventing the binding of NFkβ/p300 to the COX-2 promoter, or by binding to active sites of this enzyme, altering its activity and expression and therefore decreasing the expression of its target genes [29].

Melatonin has been shown to modulate not only aromatase, but also other enzymes involved in the local synthesis of estrogens, reducing the expression and activity of sulfatase and 17β-hydroxysteroid dehydrogenase (17βHSD) enzymes, which are involved in the formation of biologically active estrogens from inactive steroids such as androgens and estrogens. In addition, this pineal hormone increases the expression and activity of estrogen sulfotransferase (EST), increasing the number of inactive sulfoconjugated estrogens [30].

Melatonin has been reported to enhance the sensitivity of MCF-7 cells to the antiestrogenic effects of tamoxifen [31]. Furthermore, melatonin pretreatment increases the inhibition of aromatase expression, and the activity of this enzyme in cells that are exposed to the anti-steroid aminoglutethimide [32]. Additionally, melatonin also reduces or prevents the side effects of antiestrogenic therapies that are commonly used in clinic. For example, melatonin administered in animal models has been shown to reduce the hepatotoxicity induced by the aromatase inhibitor letrozole [33]. Therefore, melatonin can both improve the efficacy of conventional antiestrogens while ameliorating or eliminating their unwanted side effects [34].

## 4. Melatonin and Breast Cancer

It has been widely demonstrated that a decrease in pineal activity, due to alterations in its plasma concentration or to the circadian rhythm of secretion, induces a state of hyperestrogenism that could favour the appearance of breast tumours, due to the exposure of breast tissue to elevated estrogen levels [26].

In studies with women suffering from breast cancer, a decrease in the levels of circulating melatonin has been observed [2]. Specifically, women with tumours with positive estrogen or progesterone receptors have a lower nocturnal melatonin peak than those who have tumours with negative estrogen or progesterone receptors [35]. In addition, other authors have described lower levels of 6-sulfatexymelatonin in the urine of patients with breast cancer, along with a decrease in the amplitude of the rhythm as compared to women with benign tumours [36]. In addition, the interruption of the nocturnal increase in the production and secretion of melatonin due to excessive exposure to light during the normal dark period, for example, by light pollution or shift work, has been correlated with an increased risk of cancer [12]. This exposure alters the circadian rhythms which in turn, is reflected in the disorganization of cell clocks in peripheral tissues, deregulating these cells [12]. Most of the clinical trials with melatonin in cancer patients have been carried out by Lissoni’s group [37], and these trials demonstrate that treatment with melatonin can extend survival in patients with metastatic cancer [2].

Apart from epidemiological studies, there is also evidence from in vivo and in vitro studies. Most of the in vivo studies have been performed in animal models of mice with mammary adenocarcinomas induced by chemical carcinogens such as 7,12-dimethyl-1,2-benzanthracene (DMBA) [2] or *N*-methyl-*N*-nitrosourea (MNU) [38]. In these in vivo studies, the most general conclusion is that animals treated with melatonin or subjected to manoeuvres to enhance pineal actions show an inhibition of mammary tumorigenesis when compared to pinealectomized animals [39]. Regarding in vitro studies, melatonin reversibly inhibits the proliferation of MCF-7 cells (dependent on estrogens for their growth) through ERα [12].

### 4.1. Fat Tissue, Estrogens and Breast Cancer

Obesity is associated with an increased risk of breast cancer, especially in postmenopausal women [34]. It is therefore important to consider the obesity-inflammation-aromatase axis as a target for the treatment and prevention of breast cancer [40]. In obesity, chronic and persistent inflammation of white adipose tissue (WAT) occurs, which is associated with changes in the biology of adipocytes, leading to their dysfunction [40]. Obesity is implicated in the secretion of inflammatory factors which stimulate aromatase, which converts androgens into estrogens in adipose tissue [41]. This is the reason why an excess of adipose tissue after menopause can raise estrogen levels and increase the predisposition to breast hyperplasia and breast cancer. Furthermore, obese postmenopausal women show higher blood insulin levels, which in turn are correlated with increased breast cancer risk [34]. Insulin is capable of directly or indirectly increasing the proliferation of MCF-7 mammary epithelial tumour cells by increasing the levels of insulin-like growth factor 1 (IGF-1), which in turn regulates cell proliferation [34]. In addition, insulin decreases the synthesis of sex hormone-binding globulin (*SHBG*) that binds to estradiol in the bloodstream, thereby increasing the amount of free estradiol [34]. Many obese women have hypoadiponectinemia and hyperleptinemia due to insulin resistance, which stimulates vascular endothelial growth factor (VEGF) and nuclear factor which enhances the kappa light chains of activated B cells (NFkβ), consequently enhancing proliferation and invasion, and therefore the risk of breast cancer [34,42].

Before menopause, ovaries produce the majority of the body’s estrogen, while fat tissues produce only a small amount. In contrast, in postmenopausal women, ovarian estrogen synthesis is replaced by local synthesis in peripheral tissues. This synthesis occurs mainly in white adipose tissue (WAT) (including mammary glands), and in endothelial tissue [43]. Local synthesis of estrogens depends on the activity of different families of enzymes which transform androgens of ovarian and adrenal origin into estrogens, as well as components with weak estrogenic activity in more active forms. All of these enzymes contribute to the regulation of estrogen availability in mammary tumours [28]. The most important enzyme is aromatase, whose expression is widely distributed in extra-ovarian tissues, including subcutaneous adipose tissue, breast tissue, and cancerous breast tissue [44].

Mammary adipose tissue is made up of 90% fibroblasts, the precursors of mature adipocytes, and 10% endothelial cells. The increased incidence of breast cancer relative to the amount of adipose tissue is due in part to the fact that tumour adipose tissue has high aromatase activity, particularly in undifferentiated fibroblasts which accumulate around malignant mammary epithelial cells, reaching concentrations as much as 10 times higher than those in blood [34,45]. In fact, it has been shown that in this type of cancer, most of the activity and expression of aromatase and sulfatase are found in the fibroblasts of adipose tissue and in the vascular endothelial cells which surround the tumour, rather than in the tumour cells themselves [45]. Normal adipose tissue maintains low aromatase expression levels, mainly due to the I.4 promoter. However, in breast cancer, aromatase levels are increased by the activation of the II and I.3 promoters in malignant epithelial cells and adjacent fibroblasts, and by the I.7 promoter in adjacent endothelial cells [46,47]. Therefore, local estrogen biosynthesis in breast cancer depends on paracrine interactions between malignant epithelial cells, proximal fibroblasts, and vascular endothelial cells [48].

Some authors maintain that once epithelial cells have maligned, their growth is driven by local production of estradiol. It has been demonstrated that epithelial tumour cells secrete antiadipogenic cytokines (TNF-α, IL-11 and IL-6) and prostaglandin E2 (PGE2) which inhibit the differentiation of fibroblasts into mature adipocytes near the tumour through selective inhibition of the expression of PPARγ and C/EBPα, which are involved in adipogenic differentiation [49] (Figure 4). Furthermore, these antiadipogenic cytokines stimulate aromatase expression in these undifferentiated preadipocytes with a high capacity for estrogen synthesis around the tumour, stimulating the proliferation of tumour cells and therefore favouring tumour growth. In peritumoral mammary adipose tissue, in addition to these growth factors, positive regulation by glucocorticoids such as cortisol or dexamethasone has also been described [50]. Taken together, these processes are known as the desmoplastic reaction [48].

### 4.2. Melatonin and Desmoplastic Reaction

Melatonin modulates the desmoplastic reaction in breast cancer, reducing the production of antiadipogenic cytokines in breast tumor cells, as well as their circulating levels, inhibiting the expression of aromatase and its respective promoters, and by stimulating the two main regulators of adipogenesis, C/EBPα and PPARγ, thereby supporting the differentiation of fibroblasts into adipocytes [48,51] (Figure 4).

Conversely, it has been shown that a stimulating factor for aromatase in adipocytes is PGE2, which depends on COX activity. Melatonin has been shown to exert an inhibitory effect on cyclooxygenases, thus reducing PGE2 levels, which leads to a decrease in intracellular cAMP. Furthermore, this inhibits aromatase production by inhibiting its active promoters (I.3 and II) in peritumoral fibroblasts, which are dependent on cAMP [52]. All the foregoing will result in a reduction in the local amount of estrogens in breast tissue (Figure 4).

As already mentioned in a previous section in this review, melatonin, due to its SERM and SEEM properties, reduces estrogen levels, and could therefore be used to reduce the risk of breast cancer associated with obesity in postmenopausal women [34].

Multiple characteristics have been attributed to melatonin which correlate the administration of this hormone to a reduction in the risk of obesity related to breast cancer. These characteristics include: increased secretion of adiponectin; inhibition of aromatase expression; inhibition of effects of elevated leptin levels; reduction in blood glucose and insulin resistance, and a decrease in body fat mass [34].

## 5. Melatonin and the Estrobolome in Breast Cancer

In relation to breast cancer, one of the vital functions of gut microbiome is the regulation of steroid hormones metabolism. This is particularly relevant in the development of hormone receptor-positive breast cancer, for which circulating levels of estrogen in the body are the most important risk factor, especially in postmenopausal women. Intestinal bacteria encode enzymes capable of deconjugating the conjugated estrogen metabolites designated for excretion, pushing them back into enterohepatic circulation in a biologically active form and allowing their reabsorption [53]. Hence, an estrobolome (gut bacterial genes capable of metabolizing estrogens) enriched in deconjugating enzymes (β-glucuronidase activity [54]) promotes estrogen reabsorption and increases the total relative estrogen load, contributing to the risk of breast cancer development (Figure 5a). Furthermore, these gut bacteria also decompose polyphenols that would otherwise be indigestible and unable to induce the synthesis of short-chain fatty acids which exhibit varied estrogenic activity [53]. This could be related to STS and EST enzymes, which help to regulate the availability of estrogens in breast tumours [28]. Specifically, melatonin, as described above, reduces the expression and activity of STS, an enzyme involved in the conversion of steroids with low biological activity (sulfoconjugates), in biologically active estrogens (deconjugates), in addition to increasing the expression of EST, thereby supporting the transformation of estrogens into their inactive sulfoconjugated forms [30] (Figure 5a). Therefore, this neurohormone exerts activity which is opposite to the β-glucuronidase activity of intestinal bacteria, reducing the amount of estrogens and lowering the risk of developing breast cancer.

Bacterial composition of estrobolome in turn is probably affected by different factors (age, ethnicity, environmental influences such as diet, drinking alcohol, and the use of antibiotics) which can exert selective pressures on bacterial populations, and can cause an imbalance or dysbiosis which increases the risk of breast cancer due to elevated levels of circulating estrogens in postmenopausal women [55] (Figure 5b). Melatonin modulates the composition of the microbiota and suppresses pathogenic bacteria in the intestine due to its antioxidant activities [56]. Furthermore, significantly, enteric cells and gut microbiota produce large amounts of melatonin. Circadian disruption caused by sleep deprivation or exposure to constant light (artificial light at night–ALAN), causes an alteration in the composition of intestinal bacteria (dysbiosis) and affects the levels of melatonin in plasma and in the intestine [56]. Ren et al. demonstrated that exogenous melatonin supplementation restores microbiota composition [57] by reducing oxidative stress and the inflammatory response by suppressing TLR4 expression, all of which suggests that melatonin can interact directly with gut microbiota. Thus, since melatonin modulates microbiota composition, which is implicated in the pathogenesis of different cancers, a link exists between melatonin, microbiota, and the pathogenesis of cancer caused by dysbiosis [56] (Figure 5b).

Lower microbial richness and low microbial diversity (lower Shannon and Chao1 indices) are correlated with obesity and breast cancer risk [58,59]. In a study by Fernández et al., breast cancer patients presented a greater abundance of *Clostridiales*, *Ruminococcaceae*, *Faecalibacterium*, *Escherichia coli* and *Shigella* [59], capable of reactivating estrogens by deconjugation through their β-glucosidase and β-glucuronidase (GUS genes [53]) activity [55,60]. In addition, the Firmicutes/Bacteroidetes ratio is relevant, since an imbalance in this ratio is observed in obesity, with a greater number of Firmicutes. Therefore, dysbiosis and obesity, together with the resulting increase in circulating estrogen levels, may synergistically contribute to result in an up to 20% increased risk of breast cancer in obese women [61]. In addition, other studies have shown a higher number of Firmicutes in postmenopausal women with breast cancer [62]. In parallel, women with breast cancer and poorer sleep quality also present higher levels of Firmicutes, also indicating a possible relationship with low melatonin levels [63].

On the other hand, epigenetic modifications are frequent in tumour processes. In tumours, erroneous acetylation by histone acetyltransferase enzymes (HATs) or deacetylation by deacetylase enzymes (HDACs) occur frequently. Short chain fatty acids (SCFA) mediate some effects on the gut microbiota and the pathogenesis of some diseases, including obesity and cancer. Propionate and butyrate, as well as melatonin, increase the differentiation of preadipocytes into adipocytes by increasing the expression of C/EBPα and PPARγ [64]. The differentiating effect of butyrate on adipocytes may be partially mediated by the inhibitory activity of its histone deacetylases (HDAC), causing histone hyperacetylation [65]. This in turn inhibits the growth of cancer cells [56] by stimulating on the one hand the expression of cyclin-dependent kinase inhibitor p21, and therefore, inhibiting cell proliferation, while on the other hand, this activity is responsible for inducing cell differentiation [64], reducing aromatase activity and circulating levels of estrogens and therefore breast cancer risk.

Melatonin, as well as butyrate, regulates epigenetic modifications. It regulates the activity of SIRT1 (a class III HDAC) by modulating the cellular oxidative state [56]. In addition, melatonin has been reported to inhibit estradiol- or cadmium-induced hTERT transcription in MCF-7 cells, as well as hTERT transactivation initiated by ERα and mediated by estradiol or cadmium [21]. Finally, it should be noted that melatonin and its metabolites are structures which are similar to DNA methyltransferase inhibitors (DNMTs), thus inhibiting DNA methylation in many genes, including ARH1 in breast cancer [66].

In addition, microbiota can promote carcinogenesis by inducing chronic inflammation through the production of pro-inflammatory cytokines, by altering the balance between cell proliferation and death, and by activating innate and adaptive immune responses [59]. Specifically, WAT inflammation leads to an increase in the pro-inflammatory cytokine TNF-α, which stimulates pI.4 through the activation of NFĸβ and MAPK pathways, and therefore stimulates the expression of aromatase and its consequent estrogen production [43]. In addition, saturated fatty acids activate NFĸβ in macrophages, leading to an increase in TNF-α, IL-1β and COX2, increasing aromatase levels in preadipocytes [67].

As already discussed in this review, melatonin prevents inflammation through the reduction in the production of pro-inflammatory cytokines, thus preventing the activation of the different active promoters of aromatase and thereby reducing the amount of circulating estrogens. On the other hand, butyrate also prevents inflammation by increasing the production of IL-10 [68], thus preventing the activation of ERK1/2 mediated by TNF-α [43]. Therefore, IL-10 inhibits the expression of aromatase in breast adipose tissue induced by TNF-α, thus reducing breast cancer risk [43]. Low levels of butyrate have been observed in patients with breast cancer [53], together with elevated levels of the pro-inflammatory cytokine IL-6 [43], which can lead to cancer development and progression [69]. Microbiota could also play an important role, as studies show that mice which were fed with milk fermented with *Lactobacillus helveticus* R389 and injected with breast cancer tumour cells showed an increase in IL-10 and a decrease in IL-6 levels in serum and in mouse mammary cells, which also leads to an inhibition of breast tumour cells [70]. Therefore, a dysbiosis of the microbiota can lead to lower concentrations of butyrate and melatonin, which can result in inflammation and an increase in estrogens in the bloodstream and therefore an increased breast cancer risk [71].

Therefore, exploring variations in the composition and activity of estrobolome, as well as in melatonin levels in healthy individuals and in women with breast cancer could lead to the development of biomarkers and future targeted interventions to reduce breast cancer risk [55].

## 6. Gut Permeability, Intestinal Dysbiosis, and Circadian Disruption

Intestinal dysbiosis and disruption of the circadian rhythm are associated with various pathologies, including cancer. These disorders are associated with an increase in intestinal permeability, which allows the passage of foreign compounds to the immune system, causing inflammatory bowel diseases [72].

In circadian disruption, there is an increase in TNF-α and other pro-inflammatory cytokines, which act on the epithelial cells of the intestine, causing the loosening of tight junctions, leading to an increase in permeability [73]. In turn, this increase is associated with the dysregulation of the microbiome (dysbiosis), impacting its diversity and composition.

Therefore, a diet rich in fats and sugars that causes dysbiosis, impacts the diversity of the microbiota, favouring the appearance of diseases that are further heightened by the disruption of circadian rhythms [72]. In particular, a study by Voigt and colleagues showed that the intake of diets high in fat and sugars increased the relative abundance of the phyla *Firmicutes, Proteobacteria*, and *Verrucomicrobia*, and decreased the relative abundance of Bacteroidetes [72]. In the case of the disruption of circadian rhythms, there were no significant changes at the phylum level but there were significant changes at the family and genus levels: the phylum *Firmicutes* bacteria increased even more when combining a diet rich in fat and sugar with disruption of circadian rhythms, while the relative abundance of *Desulfosporosinus* and *Desulfotomocalum* was reduced, and *Ruminococcus* and *Sporosarcina* increased. Specifically, the most significant change observed was an increase in pro-inflammatory bacteria such as *Ruminococcus* and a reduction in *Lactobacillus*, associated with the inhibition of NFkβ. Both changes are related to increased inflammation and permeability of the intestinal barrier, which is present in certain cancers [72].

Melatonin has been shown to restore gut microbiota composition. Specifically, it reduces the relative abundance of *Clostridiales*, and increases that of *Lactobacillus*, which is correlated with a reduction in the permeability of the intestinal barrier [74].

Intestinal permeability is associated with a reduction in intestinal calcium absorption, in turn produced by a decrease in the levels of vitamin D [75], producing alterations in intestinal motility, which will be reduced and will allow the transfer of lipopolysaccharide (LPS), a component of the outer membrane of Gram negative bacteria, to the general circulation, allowing LPS to activate the pathogen recognition system CD14/TLR2/4/*MD2* [76]. The activation of this complex induces the activation of NFkβ, activating the inflammatory cytokines, TNFα, IL-1β, IL-6, COX2, and NO, and triggering strong autoimmune inflammatory activity that will eventually metastasize [6,75,76].

In addition, when TLR is activated, the serotonin transporter (SERT) is inhibited, increasing free 5*H*-serotonin. Melatonin also inhibits SERT [77]. It should be noted that high levels of LPS caused by increased intestinal permeability can suppress the synthesis of melatonin [73]. Melatonin, in concentrations similar to those obtained in the intestinal lumen after ingestion, reduces the levels of these pro-inflammatory cytokines, as well as the inhibition of the NFkβ pathway induced by bacterial LPS [73], and prevents DNA demethylation. In other words, melatonin, acting locally, can modulate inflammatory processes at the intestinal level, thereby reducing permeability [78].

Intestinal dysbiosis is associated with the suppression of the production of short-chain fatty acids (butyrates), which causes an increase in circulating LPS and an increase in intestinal permeability [71,73]. Butyrate has effects on intestinal epithelial cells, maintaining the intestinal barrier [79]. However, it can also be transferred through epithelial cells into the general circulation, where it has several effects, including inhibition of systemic immunity and the activity of the glia of the CNS. Furthermore, this short-chain fatty acid increases the cytotoxicity of natural killer cells, which are cells that deal with viruses and cancer [73]. Butyrate is also a histone deacetylase inhibitor (HDAC) and therefore a powerful epigenetic regulator, while its induction of the melatonergic pathway allows it to improve mitochondrial functioning [79]. Butyrate induces the synthesis of NAS and melatonin in the intestine, increasing the number of beneficial bacteria and strengthening the intestinal barrier [71]. Butyrate, by activating this pathway within immune cells, enables the autocrine effects of melatonin to shift activated immune cells to a quiescent state, thus producing immunosuppressive effects [73]. These effects of melatonin are mediated by an increase in the circadian gene *Bmal1*, which leads to the inhibition of pyruvate dehydrogenase kinase, which leads to the disinhibition of pyruvate in acetyl CoA, thus increasing oxidative phosphorylation (OXPHOS) and ATP of the tricarboxylic acid cycle (TCA), with acetyl CoA also being a necessary cosubstrate for arylalkylamine-*N*-acetyltransferase (AANAT), and therefore the mitochondrial melatonergic pathway, which allows melatonin to optimize mitochondrial function. Therefore, if dysbiosis occurs, butyrate levels are reduced, increasing intestinal permeability and the amount of circulating pro-inflammatory cytokines, while melatonin levels are also reduced, resulting in suboptimal functioning of the mitochondria [71].

Furthermore, given that the microbiome has diurnal fluctuations, SCFAs, including butyrate, have diurnal rhythms, and their rhythmicity could be influenced by the central circadian desynchronization of the individual, which would make the intestinal barrier even more permeable [80].

As already mentioned in the section on Trp metabolism, proinflammatory cytokines and stress, partially via dysbiosis/gut permeability, induce the synthesis of indoleamine 2,3 dioxygenase (IDO) which drives tryptophan away from the melatonergic pathway, suppressing serotonin and melatonin levels [73] and favouring the synthesis of tryptophan catabolites in the kynurenine pathway [76,81]. The metabolites derived from this pathway will later bind to AhR, exerting their different functions. Therefore, if tryptophan goes the kynurenine route instead of serotonin, NAS, and melatonin, permeability will increase [76,81]. This has been observed in various pathologies, including cancer [73].

Melatonin reduces gut permeability due to its antioxidant properties through a mitochondrial-function preservation mechanism [82]. Melatonin in turn can reduce gut permeability through the release of acetylcholine (ACh) in the vagal nerve [73], which will activate α7nAChR receptors in intestinal epithelial cells and/or mucosal immune cells [83].

Melatonin can also decrease permeability via the inflammasome, which is comprised of effectors of intestinal permeability and their interaction with intestinal bacteria. Melatonin inhibits NOD-like Receptor 3 (NLRP3) and NOD-like Receptor pyrin domain-containing- 6 (NLRP6). Both are important in regulating homeostasis and gut permeability. Specifically, NLRP6 is a regulator of murine intestinal microbiota and permeability, mediating the effects of stress induced by CRH [76]. Melatonin, by activating the α7nAChR receptor, inhibits the activation of NLRP3 by inhibiting the release of mitochondrial DNA [76]. Butyrate also acts as an NLRP3 inhibitor, prompting the observation that its effects are very similar to those of the pineal hormone [71]. Thus, permeability will vary according to several factors which regulate melatonin, such as butyrate, LPS, pro-inflammatory cytokines, and oxidative stress [73].

## 7. Clinical Trials of Melatonin in the Treatment of Breast Cancer

Breast cancer, and specifically hormone-dependent cancer, has been extensively studied in relation to melatonin. In in vitro models, it has been demonstrated that due to the SEEM and SERM properties of melatonin, it is capable of increasing the sensitivity of MCF-7 cells to the effects of tamoxifen [31], as well as to antiaromatase treatments [32]. However, there are still no clinical trials to corroborate this hypothesis.

Currently, there is only one published clinical trial in women with hormone receptor-positive breast cancer previously treated with conventional hormone therapy and to whom a melatonin supplement regime was subsequently administered. This was a randomized, double-blind, placebo-controlled trial in postmenopausal survivors of breast cancer (stages 0–III) who had completed a standard treatment with hormone therapy. The patients were treated orally, with melatonin (3 mg/day for 4 months) or placebo. The authors found no significant effect of melatonin supplementation on estradiol, IGF-1, or IGFBP-3 levels, nor on the IGF-1/IGFBP-3 ratio [84].

On the other hand, it would be interesting to investigate the possible uses of melatonin as a preventive agent for breast cancer. The association between hormone replacement therapy (HRT) and cancer risk is controversial [85]. While some clinical trials show an increased risk of breast cancer in women receiving HRT with estrogens and progesterone, others show that the risk of breast cancer after receiving HRT is reduced or insignificant [85]. Melatonin could be useful in reducing breast cancer risk after receiving HRT due to its SERM and SEEM properties. In fact, a combination of melatonin with estrogens and progesterone has been patented as a new form of HRT to reduce the possible risk of associated breast cancer [86].

Melatonin could also be used to reduce obesity-associated breast cancer risk [87] since it has been shown to prevent obesity and reduce aromatase expression and activity in animal models, thereby reducing estrogen synthesis in adipose tissue [88].

Another point of interest is the risk of breast cancer associated with exposure to environmental pollutants. In particular, an increased risk of breast cancer has been observed in women who work with chemical pollutants that have estrogenic properties (xenoestrogens). There are studies which show that both in vivo and in vitro melatonin counteracts the estrogenic effects induced by cadmium [21,89,90]. So far, there are no clinical trials assessing this property of melatonin.

Moreover, women who work at night have been shown to have an increased risk of breast cancer due to exposure to light at night [91], which inhibits melatonin secretion and induces chronodisruption [92]. This risk could be reduced if they were given a melatonin supplement.

Finally, the usefulness of melatonin as an adjuvant agent to prevent or reduce the side effects of therapies used in breast cancer has been extensively studied. In addition, a hybrid compound of melatonin and tamoxifen (*N*-desmethyl-4-hydroxytamoxifen-melatonin) has been patented (US8785501) to combine the antiestrogenic properties of both compounds and reduce the side effects of tamoxifen, such as the risk of uterine hyperproliferation [93,94].

A prospective phase II trial based on repeated measures of each patient as their own control in women with metastatic breast cancer with hormonal therapy or trastuzumab showed that melatonin improved the quality and quantity of sleep, the quality of life and social functions, lessened the severity of fatigue and increased the expression of clock genes [95]. Another randomized, placebo-controlled, double-blind study in women undergoing breast cancer surgery showed that melatonin reduced the risk of depressive symptoms [96]. Another randomized, placebo-controlled, double-blind clinical trial in postmenopausal breast cancer survivors showed that melatonin improved the quality of sleep but had no effect on hot flashes [97].

In summary, many in vivo and in vitro studies of the anticancer properties of melatonin have been described, but it would be interesting to conduct more clinical trials with melatonin to see if it really ameliorates the effects of chemotherapy and radiotherapy and prevents the side effects attendant to these therapies. Finally, it should be noted that to date there are no clinical studies investigating the relationship between the risk of breast cancer, melatonin levels, and variations in the composition of the intestinal microbiota, so it would be interesting if such studies were conducted in the future.

## 8. Possible Applications of Melatonin in Breast Cancer

It is widely described in the literature that women who work night shifts have an increased risk of breast cancer from exposure to ALAN [91], which inhibits melatonin synthesis, causing the so-called chronodisruption which we have mentioned previously [92]. If these women were to take a melatonin supplement, their risk of breast cancer associated with working at night would be reduced.

Another risk factor associated with this pathology is exposure to xenoestrogens, as may be the case in women who work in environments with chemical pollutants. Given that melatonin has been shown to counteract the estrogenic effects of Cadmium in several in vivo and in vitro studies [21,89,90], its administration to these workers should be considered in order to reduce their risk of breast cancer.

The application of melatonin as an adjuvant has been proposed in order to prevent or reduce the side effects of therapies used in breast cancer. Specifically, antiaromatase therapies induce osteoporosis which could be avoided by the ingestion of melatonin, since this indoleamine promotes the proliferation of osteoblasts and the synthesis of osteoprotegerin, thus inhibiting bone resorption and increasing bone mass [98,99]. Since melatonin has been shown to reduce the hepatotoxicity induced by aromatase inhibitors such as letrozole in vivo, the administration of melatonin in patients receiving this drug could be considered in order to reduce this side effect [13]. In addition, it has been described that tamoxifen stimulates uterine hyperproliferation, while melatonin prevents it. Accordingly, a hybrid compound has been designed that combines both molecules to prevent the side effects of tamoxifen and potentiate its antiestrogenic properties [93,94]. Regarding radiotherapy, it has been shown that applying a melatonin emulsion significantly reduced radiation-induced dermatitis in breast cancer patients [100]. In another study, Lissoni et al. tested whether melatonin lessened chemotherapy side effects in a study in patients with metastatic breast cancer with thrombocytopenia, demonstrating that melatonin prevented epirubicin-induced lowering of platelets [101].

Also, melatonin can be used to prevent the proliferation of cancer cells by reducing telomere length and the telomerase activity responsible for the development of unhealthy cells and high hormone levels, suppressing the tumour-promoting gene TP53 [102]. However, improper timing of melatonin use can result in negative outcomes, as melatonin injections in the morning can stimulate tumour growth, in the afternoon it has no effect, while in the evening it has reducing effects on cancer cell proliferation [103].

Another possible application of melatonin is to improve the quality of sleep and/or other psychological symptoms in breast cancer patients, as has been described in several studies [95,96,97,104]. At the same time, plasma and melatonin levels, as well as intestinal microbiota are affected by sleep deprivation. Nevertheless, supplementation with exogenous melatonin is able to restore the composition of the microbiota, possibly by reducing the inflammatory response and oxidative stress through the toll-like receptor-4 (TLR4)-associated signalling pathways [56]. Thus, the maintenance of the microbiota composition might be another important melatonin effect that could contribute to the prevention of breast cancer development.

On the other hand, it has been described that the microbiome has a very important role in processes such as inflammation, estrogen metabolism by estrobolome, and epigenetic alterations. Accordingly, several therapeutic and clinical applications could be proposed through genome modulation, enzymes of the designed microbiome, or especially the use of probiotics in the management of BC [105].

Specifically, one study showed that the probiotic *Lactobacillus reuteri* inhibited the early phase of carcinogenesis and increased the sensitivity to apoptosis in breast cancer cells [106]. Another study showed that the oral administration of *Lactobacillus acidophilus* had anticancer activity in mice with mammary tumours [69,107]. Another study demonstrated that mice which drank milk fermented with *Lactobacillus helveticus* R389 had elevated levels of IL-10 and decreased IL-6 levels, both in serum and mammary cells, leading to breast tumour inhibition [70]. Moreover, long-term exposure to probiotics such as *Lactobacillus casei* Shirota and soy isoflavones in Japanese females has demonstrated their chemopreventive effects on cancer development [108].

Interestingly, the use of probiotics has not been shown to change the tissue-specific microbiome, even in long-term applications. Therefore, the positive effects of probiotics in modulating the gut microbiome could prevent tumorigenesis in the breast [105].

Therefore, there is a plethora of applications for both the administration of melatonin and the regulation of the microbiota (in this case through the use of probiotics), which together could enhance the actions of each other and constitute a promising future target of study in the approach to breast cancer.

## 9. Conclusions

The objective of this review was to study the bases that support the use of melatonin as an adjuvant therapy in breast cancer due to its antiestrogenic properties. In turn, the possible bidirectional relationship between intestinal microbiota and melatonin levels is described, both impacting the development of breast cancer. Thus, both alterations in melatonin levels (circadian disruption), as well as an imbalance in the bacterial composition of the estrobolome lead to an increase in estrogen levels that promote the appearance of breast cancer.

Gut bacteria synthesize SCFAs that indirectly stimulate melatonin production through the melatonergic pathway. However, oxidative stress and pro-inflammatory cytokines stimulate the kynurenine pathway, favouring the production of tryptophan catabolites and moving tryptophan away from the melatonergic pathway, thereby reducing melatonin levels (increasing NAS in the NAS/melatonin ratio). In addition, this generates changes in gut microbiome and intestinal permeability (butyrate reduction and increase in LPS levels), increasing the inflammatory response and reducing melatonin production. All the foregoing contributes to breast cancer development.

Also, due to the antiestrogenic properties of melatonin, it reduces the expression and activity of aromatase, 17βHSD, and STS, and increases EST, affecting active and inactive estrogens levels. Furthermore, the bacterial composition of estrobolome also influences estrogen metabolism, since gut microbiome-derived β-glucuronidase activity favours the deconjugated state of estrogens and therefore increases the risk of breast cancer.

Moreover, melatonin interferes in the desmoplastic reaction by stimulating the differentiation of preadipocytes into adipocytes by increasing adipogenic cytokines (PPARγ and C/EBPα) and inhibiting antiadipogenic cytokines (IL-6, IL-11 and TNFα). Adipocytes do not express aromatase, so estrogen levels are reduced and consequently so is the risk of breast cancer.

Specifically, there are many in vivo and in vitro studies which link melatonin with breast cancer, but more clinical trials are needed to confirm the sensitizing effects of melatonin to chemotherapy and radiotherapy, included the activation of signalling linked to gut microbiota, and the prevention of side effects from these therapies.

This manuscript contributes to clarifying the relationship between melatonin and gut microbiota, representing a step in approaching breast cancer treatment. Intestinal dysbiosis and circadian disruption cause increased concentrations of estrogens, leading to the development of breast cancer. Furthermore, since dysbiosis and permeability favor the kinurenine pathway, they divert tryptophan from the melatonergic pathway, thus lowering melatonin levels. Therefore, a good balance of gut microbiota and melatonin levels are important for breast cancer prevention.

## Figures and Tables

**Figure 1 cancers-13-03141-f001:**
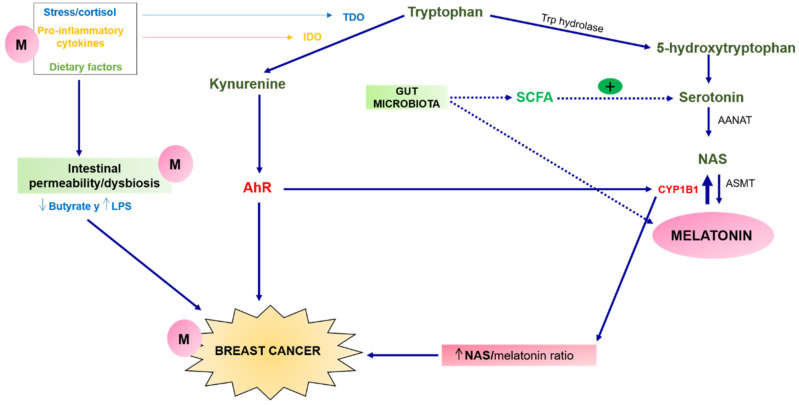
Melatonergic and kynurenine pathways. The activation of TDO by stress and cortisol, and IDO by pro-inflammatory cytokines, leads to an increase in the kynurenine pathway which implies the conversion of tryptophan into kynurenine and kynurenic acid, which can activate the AhR, leading to important changes in breast cancer cells, including a marked increase in mitochondrial CYP1B1. Mitochondrial CYP1B1 drives melatonin conversion to NAS, increasing the NAS in the NAS/melatonin ratio, which is implicated in an increased risk of breast cancer. Conversely, SCFA, produced by gut microbiota, directly or indirectly stimulates the production of melatonin by stimulating the production of serotonin which is then converted into melatonin. Gut dysbiosis and gut permeability decrease butyrate levels and increase LPS levels, favouring the appearance of breast cancer. Melatonin can counteract this effect caused by dysbiosis, preventing breast cancer. Abbreviations: AANAT: aralkylamine N-acetyltransferase; AhR; aryl hydrocarbon receptor; ASMT: acetylserotonin O-methyltransferase; CYP: Cytochrome P450; IDO: indoleamine 2,3-dioxygenase; LPS: lipopolysaccharide; M: Melatonin; NAS: N-acetylserotonin; SCFA: short-chain fatty acids; TDO: tryptophan 2,3-dioxygenase; Trp: tryptophan.

**Figure 2 cancers-13-03141-f002:**
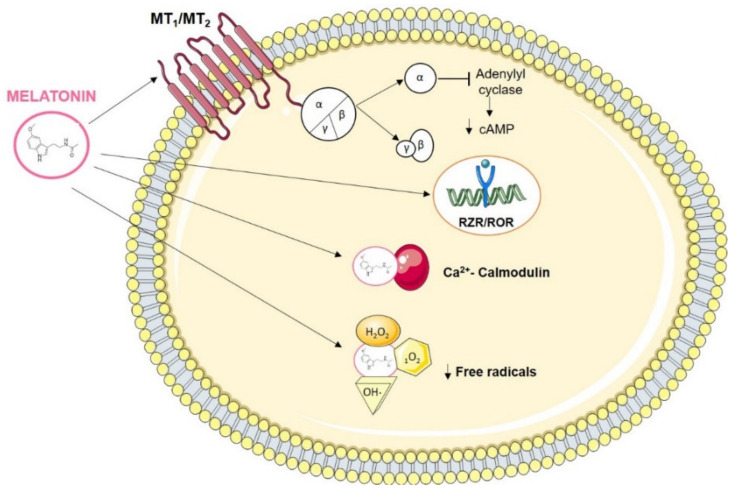
Mechanisms of action of melatonin. Melatonin can bind to its specific membrane receptors (MT1, MT2), to orphan nuclear retinoic acid receptors (RZR/ROR-α and RZR β), and interacts with calmodulin to carry out its action. Also, melatonin can act as a free radical scavenger, stimulating the expression of antioxidant enzymes and protecting cells from oxidative damage.

**Figure 3 cancers-13-03141-f003:**
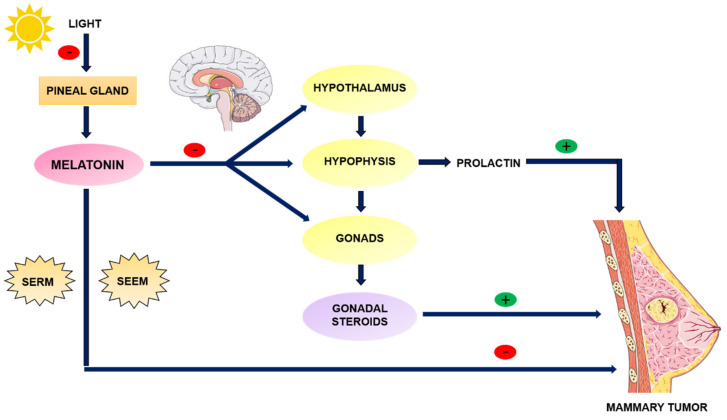
Mechanisms by which melatonin reduces the development of estrogen-mediated breast cancer. Indirect mechanism at the level of the hypothalamic-pituitary-gonad axis, with the consequent reduction of estrogenic and prolactin hormones, and melatonin’s direct antiestrogenic action at the mammary tumour cell level, acting as a SERM or as a SEEM.

**Figure 4 cancers-13-03141-f004:**
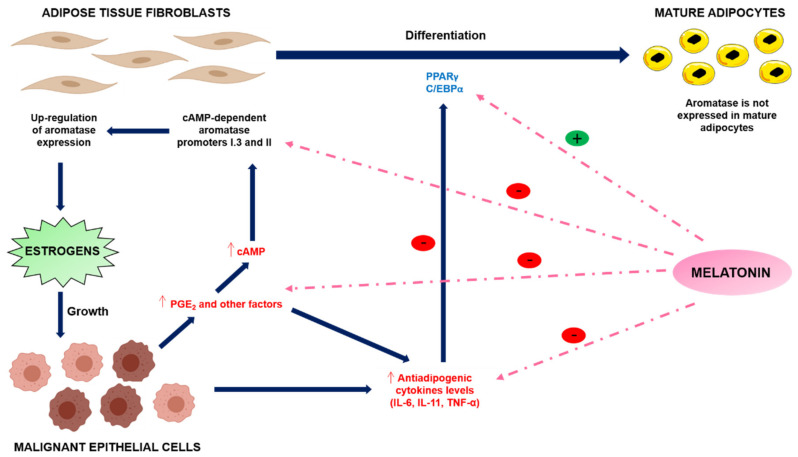
Interactions between adipose and malignant epithelial tissues: the desmoplastic reaction. Melatonin interferes with the desmoplastic reaction by stimulating the differentiation of preadipocytes into adipocytes and decreasing the aromatase activity of fibroblasts through the inhibition of the antiadipogenic cytokines expression of TNF-α, IL-6, and IL-11.

**Figure 5 cancers-13-03141-f005:**
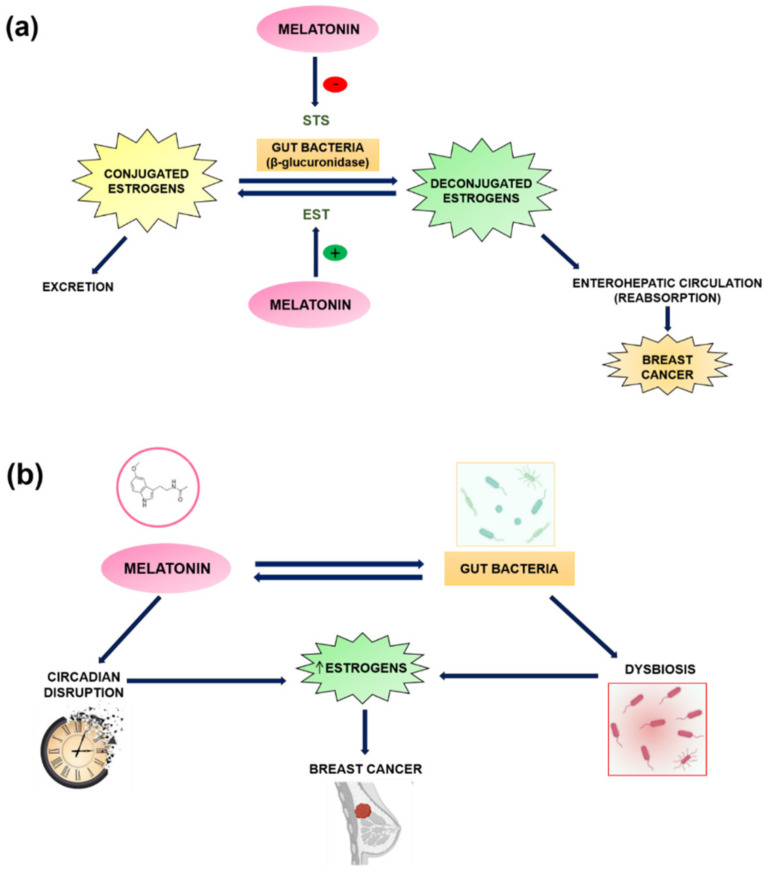
Estrogen metabolism and its relationship with melatonin, gut bacteria and breast cancer. (**a**) Estrogen metabolism. Estrogen activation through deconjugation by bacterial β-glucuronidase or by STS enzyme promotes its reabsorption and increases the risk of breast cancer. Melatonin prevents this activation of estrogens by stimulating the expression of EST enzyme, which conjugates estrogens and inactivates them, favoring their excretion, as well as by inhibition of STS. (**b**) Relationship between melatonin and microbiota. An imbalance in both melatonin (circadian disruption) and in the composition of intestinal bacteria with β-glucuronidase activity produces dysbiosis, and causes an increase in circulating estrogen levels, increasing breast cancer risk.

## Data Availability

No new data were created or analyzed in this study. Data sharing is not applicable to this article.

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
