# Peer review of "A New Paradigm in the Relationship between Melatonin and Breast Cancer: Gut Microbiota Identified as a Potential Regulatory Agent"

_cancers, 2021, doi:10.3390/cancers13133141_

Round 1

Reviewer 1 Report

The manuscript was prepared very well. The introduction section justifies the purpose of the study. I congratulate the authors for the preparation of the manuscript

However, I have the following comments:

  • It should include a brief section on Methodology
  • In section 6:

- What criteria have you used for the selection of articles?

- Why do you include revisions in line 602? You are showing essay.  Clarify this aspect.

- line 658 which references include several trials?

  • Include a brief description of what this manuscript contributes.
  • Include a section on possible applications.
  • All the figures in the manuscript: The figures are of their creation? What design program was used to make it?

Reviewer 2 Report

Manuscript entitled „ A new paradigm in the relationship between melatonin and breast cancer: gut microbiota identified as a potential regulatory agent“ is well written article. It is useful and original review that meets the scope and objectives of the journal. The review provides important insight into linkage between breast cancer and gut microbiota via regulations by melatonin. I recommend the acceptance after the implementation of following comments:

Introduction

Shortly mention also beneficial effects of gut microbiota on BC risk, excellent reference for this purpose seems: Kassayova et al. Anticancer Res. 2016 Jun;36(6):2719-28, where Lactobacillus plantarum and inulin exert prodifferentiating, antiproliferative and immunomodulatory activities, which are significantly amplified by MEL co-administration.

In the end of Introduction, please mention Aims of this study.

Chapter 2.1. All paragraphs are focused on MEL and Cancer. In this regard, correct the title of this chapter. In the second paragraph, insert one sentence/reference about MEL as synchronizer vs cancer.

Paragraph, lines 303-310, authors mentioned...Most in vivo studies...but referenced only one study  (DMBA rodent model), therefore mention also very important NMU model in this regard (MEL_BC risk_rodents), suitable reference is: Orendas et al. Int J Exp Pathol 2014 Dec;95(6):401-10.

Chapter 6, please shortly mention that there are no clinical studies regarding the linkage between MEL/BC risk and gut microbiota; because the topic of this paper is also microbiota!

Last four paragraphs of Chapter 6: MEL vs osteoblasts, MEL vs dermatitis, MEL vs thrombocytopenia, MEL vs quality of sleep are not linked with BC risk/treatment! and do not belong to this chapter/topic.

Conclusions

Last sentence, it is necessary to keep the topic of the article, please correct ... more clinical trials are needed to confirm the sensitizing effects of melatonin to chemotherapy and radiotherapy including activation of signaling linked to gut microbiota, and...

Reviewer 3 Report

This manuscript by Laborda-Illanes et al. is a generally well-written and comprehensive review in the relationship between melatonin and breast cancer. However, there are a few minor suggestions to increase reader comprehension.

  1. In 2.1 section, the authors should discuss the potential role of melatonin’s antitumour actions (cancer metastasis, apoptosis…).
  2. Figure 2 has small labeling that may not be readable.
  3. The manuscript should be checked for some minor typographical errors.

Round 2

Reviewer 1 Report

 There are no further suggestions from my side.